# Increased striatal activity in adolescence benefits learning

S. Peters[1,2] & E.A. Crone[1,2]

Adolescence is associated with enhanced striatal activity in response to rewards. This has been linked to increased risk-taking behavior and negative health outcomes. However, striatal activity is also important for learning, yet it is unknown whether heightened striatal responses in adolescence also benefit cognitive learning performance. In this longitudinal fMRI study (736 scans spanning 5 years in participants ages 8–29), we investigate whether adolescents show enhanced striatal activity during feedback learning, and whether this enhanced activity is associated with better learning performance. Here we report that neural activity indicating sensitivity to informative value of feedback peaks in late adolescence and occurs in dorsal caudate, ventral caudate, and nucleus accumbens. Increased activity in dorsal and ventral caudate predicts better current and future learning performance. This suggests that enhanced striatal activity in adolescents is adaptive for learning and may point to adolescence as a unique life phase for increased feedback-learning performance.

[1] Department of Developmental and Educational Psychology, Leiden University, 2333AK Leiden, The Netherlands. [2] Leiden Institute for Brain and Cognition, 2333ZA Leiden, The Netherlands. Correspondence and requests for materials should be addressed to S.P. (email: s.peters@fsw.leidenuniv.nl) or to E.A.C. (email: ecrone@fsw.leidenuniv.nl)

Numerous studies have demonstrated that adolescents show increased striatal activity compared to children and adults when receiving monetary rewards[1]. This increased reward-related striatal activity in adolescence has been linked to negative consequences such as risk-taking behavior and alcohol use[2,3]. However, it is possible that heightened striatal activity has not only negative, but also positive consequences in adolescence[4]. Given that adolescence is a natural transition period of increased exploration and adaption to changing environments[5], a crucial question is whether elevated striatal responses provide benefits for learning.

Much of our learning takes place by adjusting behavior following feedback, and research in adults has revealed an important role of the striatum in feedback learning[6]. The striatum consists of several subregions, such as the dorsal caudate, ventral caudate, and nucleus accumbens, with different cortical connections and functional specializations[7]. Traditionally, more dorsally located regions in the striatum are associated with cognitively complex

**Fig. 1** Mixed-model analyses for development of striatal subregions. **a** Feedback-learning task, **b** predicted trajectories for feedback-learning performance (dotted lines represent 95% confidence intervals), **c** anatomical ROIs in striatal subregions (dark blue = dorsal caudate, light blue = ventral caudate, red = nucleus accumbens), **d** predicted trajectories for sensitivity to learning signals (contrast learning > application). A quadratic age effect was the best fit for dorsal caudate (no age: Akaike Information Criterion (AIC) = 2337; linear: AIC = 2339, log-like $p = 0.606$; quadratic: AIC = 2335, log-like $p = 0.020$), ventral caudate (no age: AIC = 2370; linear: AIC = 2355, log-like $p < 0.001$; quadratic: AIC = 2339, log-like $p < 0.001$) and nucleus accumbens (no age: AIC = 2082; linear: AIC = 2056, log-like $p < 0.001$; quadratic: AIC = 2050, log-like $p = 0.004$). **e** Predicted trajectories for sensitivity to valence (contrast positive > negative learning). The best model for dorsal caudate revealed no age-related changes (no age: AIC = 2744; linear: AIC = 2746, log-like $p = 0.662$; quadratic: AIC = 2748, log-like $p = 0.487$), a linear age effect for ventral caudate (no age: AIC = 2826; linear: AIC = 2820, log-like $p = 0.007$; quadratic: AIC = 0.2819, log-like $p = 0.082$), and a quadratic age effect for nucleus accumbens (no age: AIC = 2536; linear: AIC = 2521, log-like $p < 0.001$; quadratic: AIC = 2516, log-like $p = 0.013$) ($N = 736$ scans)

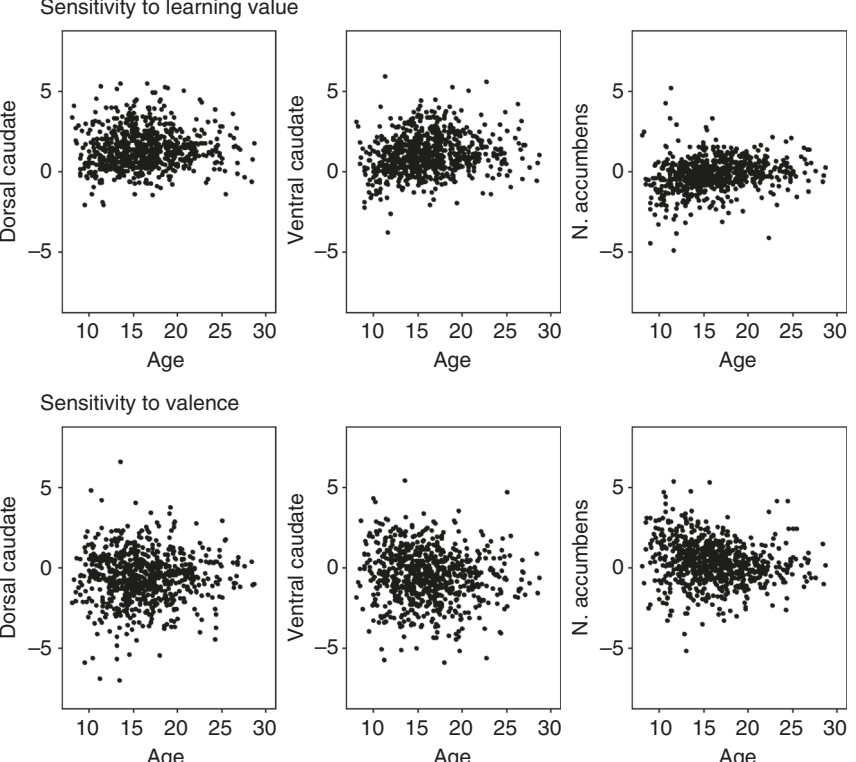

**Fig. 2** Individual data at three time points for neural activity in dorsal caudate (center-of-mass coordinates (x, y, z): 1, 7, 15), ventral caudate (0, 16, 2), and nucleus accumbens (−1, 12, −7) for sensitivity to informative value (learning > application) and sensitivity to valence (positive > negative learning) (N = 736)

instrumental learning and more ventral regions to less complex Pavlovian learning[8]. Currently little is known about how learning-related activity in these subregions changes from childhood to adulthood. This can only be tested using a longitudinal study design, as such designs allow for the investigation of within-person developmental trajectories and reduce cohort-related confounds[9].

In this study we collected functional magnetic resonance imaging (fMRI) data in a large longitudinal sample with three biannual measurement waves (age 8–25 years at the first wave). Participants performed a feedback-learning task previously found to predict reading and mathematics performance 2 years later[10]. Research in adults suggests that the striatum is sensitive to both the informative value for learning and valence of feedback[11], so we accordingly focused our analyses on the striatal response to both the informative value and the valence of feedback. The results from this study show that feedback-related activity in dorsal caudate, ventral caudate, and nucleus accumbens peaks in late adolescence/early adulthood (between ages 17 and 20) compared to children, younger adolescents, and adults. Moreover, increased activity in dorsal and ventral caudate predicts better current and future learning performance.

## Results

**Whole-brain neural activity during feedback learning**. Participants performed a feedback-learning task while fMRI images were collected. This happened in three measurement waves (total N = 736 scans, age 8–25 years at time point 1 (TP1), 51% female). On each trial, they viewed three squares with a stimulus presented below the squares. They were instructed to sort stimuli in the correct square and to use positive and negative feedback to learn the correct sorting rule for all three squares (Fig. 1a). Striatal sensitivity to learning signals was determined on a trial-by-trial

basis by comparing activity for feedback with informative value (feedback early in the learning process which was still informative for learning) and feedback without informative value (feedback during rule application, when associations were already learned). Sensitivity to valence was measured by comparing activity after positive and negative feedback during rule-learning. We first tested—on a whole-brain level—whether the striatum was indeed activated during feedback learning in our sample. We focused on the contrast learning > application (sensitivity to informative value) and positive > negative learning (sensitivity to valence), which are two different processes and may therefore have distinct developmental trajectories[12]. The whole-brain results show widespread activity including in the striatum, dorsolateral pre-frontal cortex, parietal cortex, and anterior cingulate/supplementary motor area (SMA), and survive a family-wise error (FWE)-corrected threshold at $p < 0.05$ (max T-values reached 29.31 for learning > application and 12.94 for positive > negative learning). See Supplementary Fig. 1 for the whole-brain results per contrast and Supplementary Table 1 for the brain coordinates at each time point.

**Developmental trajectories of striatal activity**. Next, we investigated the longitudinal development of neural activity in three regions-of-interest (ROIs) in the striatum (dorsal caudate, ventral caudate, and nucleus accumbens) based on the Harvard-Oxford Subcortical Atlas. Given that numerous studies ascribed different functions to dorsal and ventral caudate[7,11,13], we split the anatomical caudate into a dorsal and ventral part following recommendations by Postuma & Dagher[14] (z > 7 = dorsal) (Fig. 1c; see Supplementary Table 2 for N, mean, and SD per ROI, contrast and time point). We corrected for multiple comparisons (MC) with a Bonferroni method adjusted for correlated variables, which resulted in $\alpha = 0.027$ (Methods). Reliability over time in these

ROIs was confirmed by moderate intra-class correlation (ICC) values (Supplementary Table 3). Individual data for activity in these ROIs over three time points are presented in Fig. 2.

To investigate the longitudinal trajectory of striatal sensitivity to learning signals and valence, we used mixed-model analyses to compare a model with no age effect, a model with a linear age effect, and a model with a quadratic age effect. A linear model indicates monotonic development and a quadratic model indicates an adolescent-specific effect[15,16]. First, we focused on neural sensitivity to learning signals. The modeling results indicated that a quadratic trajectory with a peak in late adolescence (between ages 17 and 20) was the best fit for all regions. That is, the AIC value was lowest for a quadratic age effect compared to no age effect or a linear age effect in dorsal caudate (no age: AIC = 2337; linear: AIC = 2339, log-like $p = 0.606$; quadratic: AIC = 2335, log-like $p = 0.020$; $age^1$ $B = 0.54$, $age^2$ $B = -2.88$, $N = 736$), ventral caudate (no age: AIC = 2370; linear: AIC = 2355, log-like $p < 0.001$; quadratic: AIC = 2339, log-like $p < 0.001$; $age^1$ $B = 5.24$, $age^2$ $B = -5.20$, $N = 736$), and nucleus accumbens (no age: AIC = 2082; linear: AIC = 2056, log-like $p < 0.001$; quadratic: AIC = 2050, log-like $p = 0.004$; $age^1$ $B = 5.47$, $age^2$ $B = -2.95$, $N = 736$). See Fig. 1d for the predicted trajectories. For sensitivity to valence (positive > negative learning), the modeling results indicated that the best model for dorsal caudate was one with no age-related changes (no age: AIC = 2744; linear: AIC = 2746, log-like $p = 0.662$; quadratic: AIC = 2748, log-like $p = 0.487$; $N = 736$). For ventral caudate, there was a linear decrease with age for positive > negative learning (no age: AIC = 2826; linear: AIC = 2820, log-like $p = 0.007$; quadratic: AIC = 2819, log-like $p = 0.082$; $age^1$ $B = -4.63$, $N = 736$). For nucleus accumbens, the best model was a quadratic decrease with age in nucleus accumbens activity for positive > negative learning (no age: AIC = 2536; linear: AIC = 2521, log-like $p < 0.001$; quadratic: AIC = 2516, log-like $p = 0.013$; $age^1$ $B = -5.80$, $age^2$ $B = 3.40$, $N = 736$). See Fig. 1e for the predicted trajectories.

**Striatal activity and learning performance**. Next, we tested whether enhanced striatum activity was linked to better learning performance. Feedback-learning performance was defined as the percentage of feedback during the learning phase, which was successfully used on the next trial (stay for positive feedback, switch for negative feedback). The analyses were performed without behavioral outliers ($N = 5$; Supplementary Fig. 2) to ensure we included only participants who understood the task. First, we used mixed-model analyses to assess the developmental trajectory for learning performance by comparing a model with no age effect, a model with a linear age effect and a model with a quadratic age effect. The results indicated that the model with a quadratic age effect (peaking around late adolescence/early adulthood around age 20–21) was the best fit (no age: AIC = 4371; linear: AIC = 4287, log-like $p < 0.001$; quadratic: AIC = 4250, log-like $p < 0.001$; $age^1$ $B = 47.84$, $age^2$ $B = -30.12$, $N = 731$). See Fig. 1b for the predicted trajectory.

Subsequently, we tested the relationship between striatal activity and learning performance in two ways: (i) using mixed-model analyses, we tested whether learning performance was predicted by striatal activity at three time points, including within-subject changes in learning performance and within-subject changes in striatal activity over time, and (ii) using regression analyses, we tested whether striatal activity had predictive value for current learning performance and for performance 2 or 4 years later. First, we used mixed-model hierarchical regression analyses to test whether learning performance was best predicted by a model with a quadratic age effect

**Table 1 Predicting learning performance above a quadratic effect of age from neural sensitivity to informative value (learning > application) (N = 731)**

| Model parameter | B | SE | t | p-value |
|---|---|---|---|---|
| Intercept | 94.62 | 0.21 | 458.31 | < 0.001 |
| Age¹ | 48.90 | 5.13 | 9.54 | < 0.001 |
| Age² | −29.86 | 4.74 | −6.30 | < 0.001 |
| Dorsal caudate | 0.51 | 0.16 | 3.09 | 0.002 |
| Intercept | 94.62 | 0.21 | 458.00 | < 0.001 |
| Age¹ | 48.50 | 5.13 | 9.46 | < 0.001 |
| Age² | −29.87 | 4.75 | −6.30 | < 0.001 |
| Ventral caudate | 0.44 | 0.16 | 2.71 | 0.007 |

Dorsal and ventral caudate predicted learning performance. When adding striatal activity to the model, we used the regression residuals from the best age model for that region

alone, or whether adding striatal activity resulted in a better prediction of learning performance. When adding striatal activity to the model, we used regression residuals from the best age model for that region, e.g., when adding nucleus accumbens activity, we added the regression residuals for nucleus accumbens predicted by a quadratic age effect. The results indicated that for sensitivity to learning value, learning performance was predicted over $age^2$ by dorsal caudate ($B = 0.51$, $p = 0.002$, $N = 731$) and ventral caudate ($B = 0.44$, $p = 0.007$, $N = 731$), such that higher sensitivity to learning signals was associated with better learning performance (see Table 1 for the regression parameters and Fig. 3a for an illustration of the longitudinal relation between learning performance and striatal activity). Nucleus accumbens activity did not predict behavioral performance ($B = 0.19$, $p = 0.257$, $N = 731$). For sensitivity to valence, better performance was predicted by more activity for negative > positive learning in ventral caudate ($B = -0.34$, $p = 0.036$, $N = 731$, but note that this result does not survive a Bonferroni MC correction threshold adjusted for correlated variables; see Supplementary Table 4 for regression parameters). Better performance was not predicted by more activity for negative > positive learning in either the dorsal caudate ($B = -0.21$, $p = 0.183$, $N = 731$) or nucleus accumbens ($B = -0.20$ $p = 0.217$, $N = 731$).

We performed additional mixed-model hierarchical regression analyses excluding participants who scored 100% on learning performance ($N = 15$ at TP1, $N = 27$ at TP2, and $N = 69$ at TP3; resulting in $N = 620$), because the task was designed to be relatively easy to strike a balance between young children achieving sufficient learning success and adults not reaching ceiling performance. The mixed-model hierarchical regressions indicated that sensitivity to informative value still predicted learning performance over $age^2$ in dorsal caudate ($B = 0.67$, $p < 0.001$, $N = 620$) and ventral caudate ($B = 0.50$, $p = 0.005$, $N = 620$). Sensitivity to valence no longer predicted learning performance over $age^2$ for negative > positive learning in ventral caudate ($B = -0.34$, $p = 0.058$, $N = 620$).

We also explored whether striatal activity could be used to predict future learning performance 2 and 4 years later. Regression analyses showed that for sensitivity to informative value, TP1 striatum activity predicted TP2 learning performance 2 years later in dorsal caudate ($\beta = 0.19$, $p = 0.006$, $N = 211$) and ventral caudate ($\beta = 0.19$, $p = 0.005$, $N = 211$). TP2 striatum activity also predicted TP3 learning performance in ventral caudate ($\beta = 0.19$, $p = 0.010$, $N = 195$) but not in dorsal caudate ($\beta = 0.14$, $p = 0.056$, $N = 195$) and nucleus accumbens ($\beta = 0.14$, $p = 0.053$, $N = 195$). TP1 striatum activity could not predict TP3 learning performance 4 years later. For sensitivity to valence, TP1 striatum activity did not predict future TP2 or TP3 learning

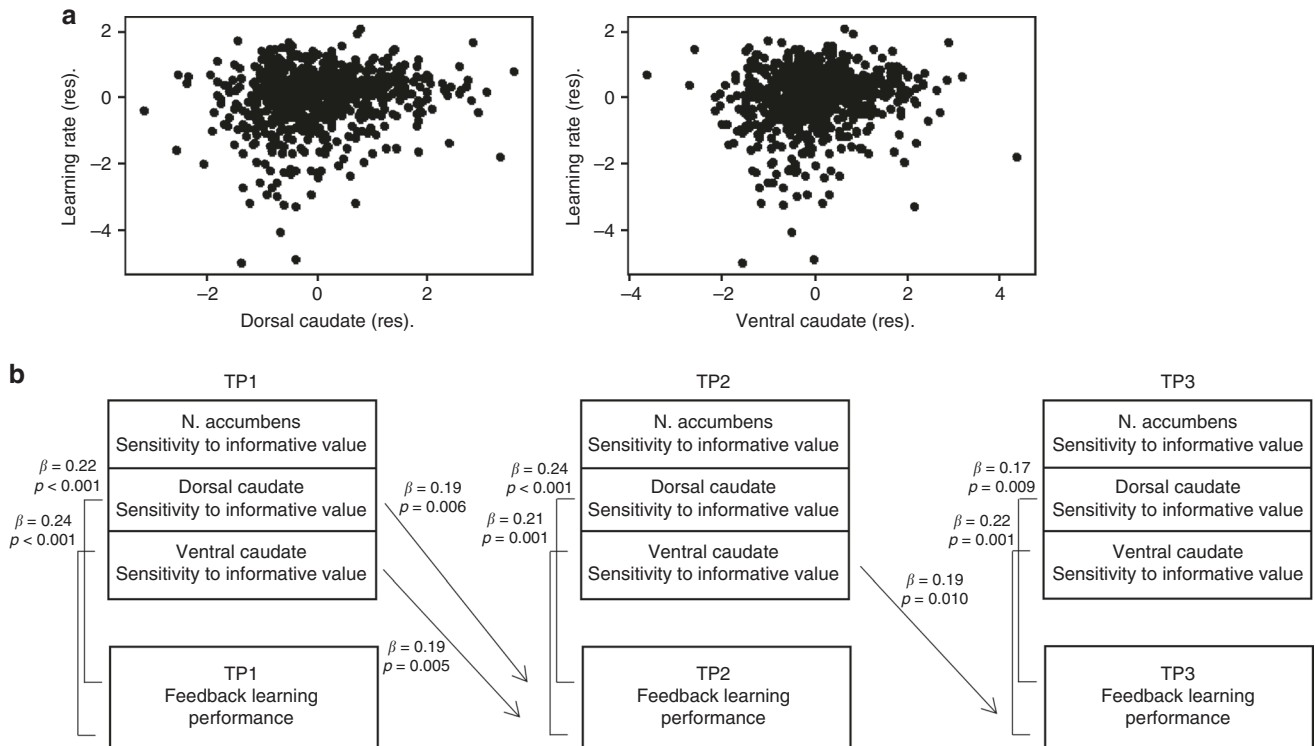

**Fig. 3** The relation between striatal activity and learning performance. **a** Scatterplot of the relation between learning performance and sensitivity to informative value in dorsal caudate ($B = 0.51$, $p = 0.002$) and ventral caudate ($B = 0.44$, $p = 0.007$) ($N = 731$). The regression residuals for learning performance predicted by $age^2$ and dorsal/ventral caudate predicted by $age^2$ are plotted. **b** Schematic overview of the cross-sectional and longitudinal regression analyses demonstrating that sensitivity to informative value in dorsal and ventral striatum activity predicts learning performance at the same time point. Activity in dorsal and ventral caudate predicts learning performance 2 years later

performance. See Fig. 3b for an overview of the significant cross-sectional and longitudinal regression analyses.

**Developmental trajectories of cortical activity.** One of the mechanisms through which this adolescent-specific increased striatal sensitivity to informative value might benefit learning performance is by increasing the recruitment of cognitive control regions in the frontoparietal network. These regions were also involved during this feedback-learning task (Supplementary Fig. 1)[17]. To test whether frontoparietal activity also showed an adolescent peak for sensitivity to informative value, we extracted ROI values from anatomical ROIs (Harvard-Oxford Cortical Atlas) in middle frontal gyrus (MFG), anterior cingulate cortex (ACC), SMA, and superior parietal lobule (SPL) (Fig. 4a). Note that data from these regions for the first two time points have been reported earlier[17]. Longitudinal mixed-model analyses were used to compare a model with no age effect, a linear age effect, and a quadratic age effect. The analyses revealed that, similar to the striatum, a quadratic trajectory (peaking in late adolescence/ early adulthood between ages 16–20) was the best fit for MFG (no age: AIC = 2649; linear: AIC = 2651, log-like $p = 0.841$; quadratic: AIC = 2643, log-like $p = 0.001$; $age^1$ $B = -0.82$, $age^2$ $B = -5.01$, $N = 736$), SMA (no age: AIC = 3017; linear: AIC = 3012, log-like $p = 0.010$; quadratic: AIC = 3009, log-like $p = 0.020$; $age^1$ $B = 5.50$, $age^2$ $B = -4.70$, $N = 736$), and SPL (no age: AIC = 2717; linear: AIC = 2717, log-like $p = 0.155$; quadratic: AIC = 2710, log-like $p = 0.004$; $age^1$ $B = -2.95$, $age^2$ $B = -4.80$, $N = 736$). ACC, however, showed no age-related changes (no age: AIC = 2441; linear: AIC = 2442, log-like $p = 0.403$; quadratic: AIC = 2443, log-like $p = 0.416$; $N = 736$). See Fig. 4b for the predicted developmental trajectories. Mixed-model regression analyses revealed that enhanced activity in MFG ($B = 0.45$, $p = 0.007$) and SPL ($B = 0.46$,

$p = 0.007$) predicted better learning performance over a quadratic age effect alone.

## Discussion

Taken together, the results presented here show that striatal activity indicating heightened sensitivity to learning signals peaks in late adolescence, and that enhanced striatal activity can predict better current and future learning performance. These findings are based on a large-scale longitudinal design, which is crucial for investigating developmental trajectories because the results are based on within-person changes over time and are robust to inter-individual differences.

Adolescents showed elevated striatal activity for feedback when they learned new rules compared to feedback that occurred when they applied known rules. The degree to which the striatum—including dorsal caudate, ventral caudate, and nucleus accumbens—was sensitive to the informative value of the feedback was highest in late adolescence (between ages 17 and 20) and predicted better learning performance. Similarly, neural activity in cortical regions including the MFG, parietal cortex, and SMA also showed a peak in activity related to informative value around this age (between 16 and 20). Together with the behavioral results, which demonstrated a peak in feedback-learning performance in late adolescence/early adulthood, our data suggest that this developmental phase may be an optimal period for feedback learning. Enhanced learning performance in adolescence has been reported in prior studies[18–20] but has not yet been related to striatal activity in a longitudinal study.

Interestingly, research has also suggested that risk-taking is most prevalent in late adolescence and early adulthood[21]. Possibly, late adolescence is an adaptive life period during which the brain is optimally responsive to learning signals and new

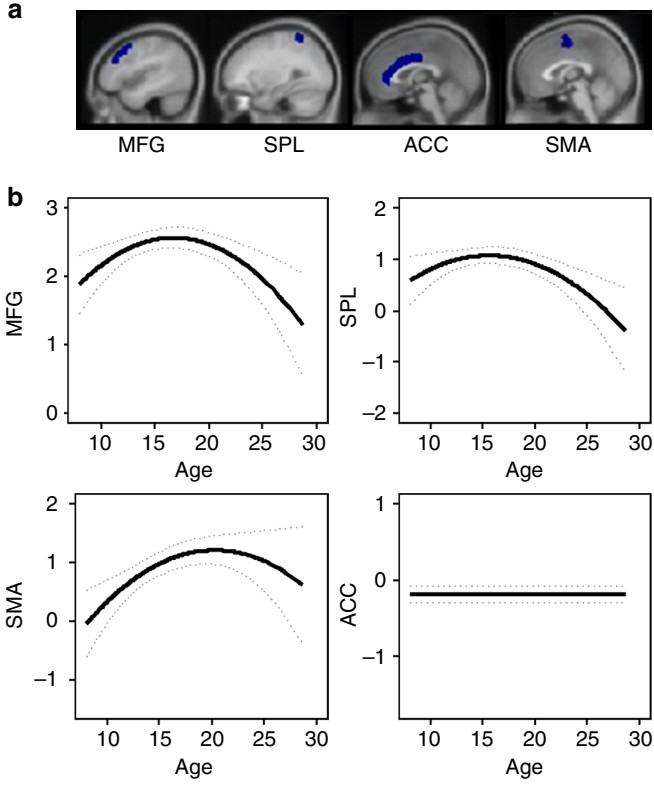

**Fig. 4** Mixed-model analyses for development of cortical regions in the frontoparietal network. **a** Anatomical ROIs (Harvard-Oxford Cortical Atlas). Center-of-mass coordinates were (x, y, z): middle frontal gyrus (MFG) (−4, 22, 43); superior parietal lobule (SPL) (7, −48, 63); supplementary motor area (SMA) (0, −2, 58); anterior cingulate cortex (ACC) (1, 20, 24). **b** Predicted developmental trajectories for sensitivity to learning signals. A quadratic age effect was the best fit for MFG (no age: AIC = 2649; linear: AIC = 2651, log-like $p$ = 0.841; quadratic: AIC = 2643, log-like $p$ = 0.001), SMA (no age: AIC = 3017; linear: AIC = 3012, log-like $p$ = 0.010; quadratic: AIC = 3009, log-like $p$ = 0.020), and SPL (no age: AIC = 2717; linear: AIC = 2717, log-like $p$ = 0.155; quadratic: AIC = 2710, log-like $p$ = 0.004). ACC showed no age-related changes (no age: AIC = 2441; linear: AIC = 2442, log-like $p$ = 0.403; quadratic: AIC = 2443, log-like $p$ = 0.416) (N = 736)

after negative compared to positive feedback across ages (8–29 years), which fits with research linking activity in dorsal caudate to punishment-based learning[25]. Ventral caudate and nucleus accumbens activity became stronger for negative relative to positive feedback with increasing age. This is in accordance with prior studies showing age-related neural changes in performance monitoring, such as larger event-related negativity in scalp responses after errors[26]. There was relatively more activity after positive feedback in the nucleus accumbens in childhood, whereas in early adulthood there were no longer valence differences in activity. This finding is similar to prior research in adults[27], and suggests that the nucleus accumbens responds to cognitive learning signals regardless of whether they are presented as positive or negative feedback. However, this is not true for the ventral caudate, which showed increased responses to negative compared to positive feedback. These increased responses were associated with better learning performance. Taken together, these findings provide evidence for dissociable contributions of striatal subregions to learning and underline the importance of considering these subregions individually.

An important remaining question is how the striatum connects to other brain regions to support enhanced learning performance. Potentially, enhanced striatal activity leads to an upregulation of cognitive control regions, such as the MFG and parietal cortex, and consequently an increase in cognitive performance[28]. In agreement with this idea, we found that regions in the fronto-parietal network—including the MFG, SMA, and SPL—also showed a late adolescent peak (between ages 16 and 20) in sensitivity to informative value. However, as we did not perform causal connectivity analyses, we cannot identify whether or not this is the exact mechanism by which these effects occur. In addition, different methods such as probabilistic learning tasks and computational approaches could further our understanding of learning differences between adolescents and adults[18,29,30]. Other important remaining questions are whether these findings are similar for more complex cognitive tasks, as well as whether they are relevant for school learning and whether there are differences between supervised and unsupervised learning across development[10,31]. When interpreting our findings, it should be taken into account that, as is inherent with longitudinal designs, there may be a learning effect due to performing the task repeatedly. However, because we used an accelerated longitudinal design (with different starting ages) rather than following a single cohort, there is overlap between age and wave (i.e., practice), thereby we control for possible age differences in practice effects over sessions[32].

In conclusion, our findings make an important contribution to theoretical models of adolescence. Classic dual-systems models of adolescent development emphasized vulnerabilities of the adolescent brain due to heightened affective responses combined with immature cognitive control[33,34]. Our results support more recent theories, which expand upon classic models and emphasize the intertwined nature of cognitive and affective brain systems in adolescence[5,23]. For instance, increased activity in motivational brain regions such as the striatum may result in increased recruitment of the control system depending on motivational salience. This study provides a neurobiological explanation for this effect by demonstrating that enhanced striatum sensitivity may underlie an adolescent-specific window of opportunity for feedback learning.

environments, which may explain both of these findings. This idea fits with prior research showing that enhanced exploration and novelty seeking are not unique to adolescent humans, but are also present during adolescence in other mammalian species (such as rats and mice)[22]. Our results also resonate with recent theories of adolescent development that emphasize adaptive flexibility, highlighting that adolescence is not only a period of negative health consequences, but also a unique period for exploration and adaptation[5,23].

With regard to potential specific functions of striatal sub-regions, we found that dorsal caudate, ventral caudate, and nucleus accumbens all showed an adolescent peak (between ages 17 and 20) in neural responses to feedback with informative value for learning. While increased activity in dorsal and ventral caudate was linked to better learning performance, activity in nucleus accumbens was not. Dorsal and ventral caudate activity also predicted learning performance 2 years later, marking the first steps toward using striatal activity to investigate future learning potential[24]. Aside from sensitivity to informative value of feedback, we also investigated how neural reactions to positive and negative feedback change across development. These analyses showed that dorsal caudate remained consistently more active

## Methods

**Participants.** At the first time point (TP1), a total of 299 participants between ages 8 and 25 years participated in this MRI study. About 28 participants were excluded for the following reasons: $N$ = 4 did not complete MRI session, $N$ = 1 disclosed ADD diagnosis, $N$ = 22 movement > 3 mm, $N$ = 1 reported medicine use. In total,

data from 271 participants were included at TP1. At TP2 2 years later, 254 participants were scanned (most dropout was due to braces, $N = 33$). Of these 254 participants, there were several exclusions: $N = 12$ movement > 3 mm, $N = 5$ scanner artifact, $N = 2$ data processing error, $N = 1$ disclosed ADD diagnosis, $N = 1$ reported medicine use. In total, 233 participants were included at TP2. At TP3, 2 years after TP2, 243 participants were scanned (dropout due to braces: $N = 11$). Of these, several were excluded: $N = 3$ did not complete MRI session, $N = 4$ movement > 3 mm, $N = 2$ data processing error, $N = 1$ disclosed ADD diagnosis, $N = 1$ reported medicine use. In total, 232 participants were included at TP3. The grand total of included data was 736 scans.

Participants were not allowed to participate if they reported current use of psychotropic medication or a psychiatric diagnosis. IQ was estimated at the first two time points with two subtests of the WISC-III (participants under 16; TP1: $N = 195$; TP2: $N = 119$) or WAIS-III (participants 16 and older; TP1: $N = 76$; TP2: $N = 114$): Similarities and block design at TP1, and vocabulary and picture completion at TP2. Estimated IQ scores were within the normal range at both TP1 (80–143, $M = 110.00$, SD = 10.34, $N = 271$) and TP2 (80–147.50, $M = 108.36$, SD = 10.44, $N = 233$). IQ was not related to age, suggesting no intelligence differences across our ages (Pearson's correlation $r = −0.74$, $p = 0.200$ at TP1, $N = 271$). All participants and their parents (for participants < 18 years) provided written informed consent and the study was approved by the Medical Ethical Committee at Leiden University Medical Center. Children received presents for their participation and their parents received payment meant for travel costs. Adult participants (≥18 years) received payment for participating in the study. All anatomical scans were reviewed by a radiologist and no clinically relevant abnormalities were reported.

**Feedback-learning task**. Participants performed a feedback-learning task while fMRI images were collected. Data on frontoparietal activity from the first two time points have been published in prior studies[17,35]. On each trial, participants viewed three squares with a stimulus presented underneath. Participants were instructed to use performance feedback to determine which stimulus belonged to which square (Fig. 1a). Performance feedback was presented after each choice, as a minus sign for negative feedback and a plus sign for positive feedback. After 12 trials, or when a criterion was reached (placing each of the three stimuli in the correct box at least two times), a new sequence with three new stimuli was presented. This criterion was used to strike a balance between the number of trials in the learning and the application phase. The total number of sequences for TP1 and TP2 was 15 sequences, resulting in a maximum total of $15 \times 12 = 180$ trials per participant. For TP3, 10 sequences (max. 120 trials) were presented due to time constraints. All analyses were also performed with only the first 10 sequences for TP1 and TP2, which resulted in highly similar findings. Before the MRI session, participants practiced the task for three sequences. The time line of trials was as follows: fixation cross (500 ms), stimulus + response (2500 ms), feedback (1000 ms). Inter-trial intervals were jittered and optimized using Optseq, with intervals varying between 0 and 6 s.

**FMRI analyses**. For the fMRI contrasts, we focused on the contrasts sensitivity to informative value (learning > application; comparing feedback early in the learning process which is still valuable for learning, with feedback for associations that were already learned), and sensitivity to valence (positive > negative learning; comparing positive and negative feedback, but only for feedback that is still valuable for learning). To determine these feedback types, we distinguished between a "learning phase" and an "application phase" using an individual trial-by-trial approach for each stimulus. To determine whether a stimulus was in the learning or application phase, we used trial-by-trial analyses and took into account future responses on the next trial this stimulus was presented. Specifically, stimuli were classified into the learning phase when participants had not yet provided the correct location for this stimulus in prior trials, and where still using the feedback to determine the correct location (i.e., the feedback still had informative value for learning). Trials during the learning phase which did not result in learning, i.e., the trials where the feedback was not successfully used on the subsequent trial ("switch" for negative feedback, "stay" for positive feedback; at TP1: $M = 5.83$, SD = 7.39, $N = 271$) were excluded from further analyses. Stimuli were classified into the application phase when the stimulus was already sorted correctly in a previous trial and continued to be sorted correctly on the next trial. Therefore, in the fMRI contrast learning > application we focused on neural regions distinguishing between feedback that is valuable for learning vs. not valuable for learning (i.e., neural sensitivity to informative value).

For the fMRI contrast positive > negative learning, we could investigate valence effects while controlling for informative value. That is, we contrasted only positive and negative feedback during the learning phase (i.e., we excluded feedback presented after an association had already been learned, because this may rely on very different processes compared to feedback earlier in the learning process[36]. At TP1, an average of 69.65 trials (SD = 5.51, $N = 271$) was classified as learning phase trials (of which positive: $M = 41.50$, SD = 4.02, $N = 271$; of which negative: $M = 28.15$, SD = 7.56, $N = 271$), and an average of 59.92 trials (SD = 6.56, $N = 271$) was classified as application phase trials. In total, participants needed an average of 139.16 trials (SD = 9.98, $N = 271$) to complete the task.

To measure feedback-learning performance, we calculated the percentage of trials in the learning phase for which feedback was successfully used on the next trial (stay for positive feedback, switch for negative feedback), compared to the total number of trials during the learning phase. Extreme behavioral outliers were excluded for analyses with learning performance (Supplementary Fig. 2, $N = 5$). This resulted in $N = 268$ at TP1 (learning performance $M = 93.43$, SD = 5.11), $N = 231$ at TP2 (learning performance $M = 94.70$, SD = 4.67) and $N = 232$ at TP3 (learning performance $M = 96.05$, SD = 4.72). Feedback-learning performance showed a weak age-controlled correlation with IQ, which was measured at TP1 and TP2 (TP1: Pearson's $r = 0.140$, $p = 0.022$, $N = 271$; TP2: Pearson's $r = 0.155$, $p = 0.019$, $N = 233$). There were no sex differences in feedback-learning performance at all three time points (according to independent samples $T$-tests, all $p$s > 0.196). Learning performance showed reliability over time (ICC = 0.494, $N = 731$).

**MRI data acquisition**. We used the same Philips 3T MRI scanner and settings for all time points. The following settings were used: TR = 2200 ms, TE = 30 ms, sequential acquisition, 38 slices, slice thickness = 2.75 mm, field of view (FOV) = $220 \times 220 \times 114.68$ mm. We acquired a high-resolution 3D T1-FFE anatomical scan after the experimental task (TR = 9.8 ms, TE = 4.6 ms, 140 slices, voxel size = $0.875 \times 0.875 \times 1.2$ mm, FOV = $224 \times 177 \times 168$ mm, flip angle = 8). Prior to the MRI scan, participants were placed in a mock scanner to accustom them to the MRI environment and noise.

**FMRI data analysis**. We performed whole-brain analyses using SPM8 (Wellcome Department of Cognitive Neurology, London). The following preprocessing steps were used: slice timing correction, realignment (motion correction), normalization, and smoothing (6 mm FWHM isotropic Gaussian kernel). T1 templates were based on the MNI305 stereotaxic space[37]. The task was an event-related design and the events (positive learning, negative learning, and application) were time-locked with 0 duration to the moment of feedback presentation. Motion regressors were added to the model. All other trials (i.e., trials that did not result in learning or too-late trials) were modeled as events of no interest. These events were used as covariates in a general linear model together with a set of cosine functions that high-pass filtered the data. The least-squares parameter estimates of height of the best-fitting canonical HRF for each condition were used in pair-wise contrasts. The main fMRI contrasts were learning > application (sensitivity to informative value) and positive > negative learning (sensitivity to valence). The contrast images were submitted to higher-level group analyses. Whole-brain fMRI analyses were performed with an FWE voxel-level corrected threshold at $p < 0.05$.

**Region-of-interest analyses**. ROI analyses were performed with the MarsBaR toolbox (v. 0.42) in SPM8[38]. ROIs (caudate nucleus and nucleus accumbens) were based on the Harvard-Oxford Subcortical Atlas (thresholded at 50%). Numerous studies have suggested differential functional specialization of dorsal and ventral caudate[7,11,13], therefore we followed recommendations by Postuma & Dagher[14] and split the caudate ROI into a ventral and dorsal part ($z > 7$ = dorsal). We averaged across left and right hemispheres. Center-of-mass coordinates were $(x, y, z)$: dorsal caudate $(1, 7, 15)$; ventral caudate $(0, 16, 2)$; nucleus accumbens $(−1, 12, −7)$. Mean and SD values for each contrast, each ROI, and each time point are reported in Supplementary Table 2. In the follow-up analyses, we tested for developmental trajectories in several cortical regions: MFG, SPL, SMA, and ACC, using atlas-based ROIs (Harvard-Oxford Cortical Atlas; thresholded at 50%). Center-of-mass coordinates were $(x, y, z)$: MFG $(−4, 22, 43)$; SPL $(7, −48, 63)$; SMA $(0, −2, 58)$; ACC $(1, 20, 24)$.

**Statistical analyses**. We used mixed-model analyses to test the shape of developmental trajectories for behavior and neural activity[16] using the NLME package in R[39]. With this package, it is possible to test for fixed effects (effects that are similar for all participants) and random effects (effects that vary across participants) of age on brain activity. Models were compared using the Akaike Information Criterion (AIC; lower AIC values indicate a better fit of the model to the data) and we additionally tested with log-likelihood tests whether changes in model fit were large enough to be significant. We first tested for each ROI which shape best described the developmental trajectory. We started with a base model, which included a fixed intercept and a random intercept to account for the repeated nature of the data. This base model was tested against three models to test the shape of the grand mean trajectory for age. We tested for a linear age effect (monotonic development) and a quadratic effect of age (an adolescent-specific effect) by adding polynomial functions for age to the base model[15,16]. After determining the developmental trajectory for age and neural activity, we furthermore tested whether neural activity could predict feedback-learning performance over age alone. We started with the best-fitting age model (quadratic) for behavioral performance ROI and tested whether a model with both age and activity in the ROI resulted in a better model fit compared to age alone. Results were corrected for MC using a Bonferroni method adjusting for correlated variables (http://www.quantitativeskills.com/sisa/calculations/bonfer.htm)[40,41]. The average correlation between variables (three ROIs, two contrasts, separately per contrast) was $r = 0.66$, which resulted in an adjusted significance level (2-sided adjusted) of $\alpha = 0.027$. We reported when analyses were significant at $p < 0.05$ but did not survive correction for MC.

**Data availability**. All relevant data are available from the authors upon reasonable request.

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

## Acknowledgements

This research was funded by a starting grant of the European Research Council (ERC-2010-StG-263234 awarded to E.A.C.) and a grant from the Netherlands Organization for Scientific Research (NWO-VICI 453-14-001 awarded to E.A.C.). We would like to thank Laura van der Aar, Sibel Altikulaç, Neeltje Blankenstein, Barbara Braams, Suzanne van de Groep, Juliette Cassé, Dianne van der Heide, Jorien van Hoorn, Cédric Koolschijn, Babette Langeveld, Kyra Lubbers, Batsheva Mannheim, Mara van der Meulen, Rosa Meuwese, Sandy Overgaauw, Jiska Peper, Elisabeth Schreuders, Merel Schrijver, Jochem Spaans, Marije Stolte, Erik de Water, and Bianca Westhoff for their help with data collection, Ferdi van de Kamp for help with data analyses, and Anna Van Duijvenvoorde and Berna Güroğlu for carefully reviewing the manuscript. Finally we would like to thank all participants and their parents for their collaboration.

## Author contributions

S.P. and E.A.C. designed the research, S.P. collected the data, S.P. and E.A.C. analyzed the data, S.P. and E.A.C. wrote the manuscript.

## Additional information

**Competing interests:** The authors declare no competing financial interests.

]

