## [Peer Review File · Nature Communications]

Editorial Note: this manuscript has been previously reviewed at another journal that is not operating a transparent peer review scheme. This document only contains reviewer comments and rebuttal letters for versions considered at *Nature Communications*, which is why there are comments from Reviewer 2 only. Mentions of prior referee reports have been redacted.

Reviewers' comments:

Reviewer #2 (Remarks to the Author):

The goal of this study was to test the hypothesis that enhanced striatal activation in adolescents benefits learning performance. It is an extremely well-powered study (n=736 scans) that used a longitudinal design and the findings are interesting. I have very few comments, most of are clarifications or pertain to concerns raised in the previous round of review [redacted].

Methods

How often was it the case that performance criterion was not reached (e.g. the participant got the full 12 trials before moving onto a new sequence)?

for the 'sensitivity to learning value' contrasts, did you exclude the sequence trials in which the participant did not learn (eg. they needed the full 12 trials before moving on)?

Since performance criterion was defined as placing each stimuli in the correct box twice, how do you know the second time in the correct place wasn't just by chance?

Did these data control for IQ?

Results: In response to my question about how the learning>application contrast yielded a quadratic trajectory whereas the positive>negative learning contrast did not, the authors have explained that these two contrasts measured two different processes. This needs to be clarified and explained in the text.

Also, further discussion of why the relative peak in accumbens activity should be considered relevant when there is no difference between learning and application should be included. This discussion should include the references the authors provided to show that the nucleus accumbens can be blind to differences in valence, i.e. shows a similar response to positive and negative feedback.

Discussion

Some of the Discussion is a bit overstated in terms of causality. These data cannot tell us whether the unique adolescent brain patterns of activation lead to better learning, as this sentence suggests: "adolescent-specific sensitivities in brain development provide a window of opportunity for learning"

Relatedly, this sentence suggest big leaps that should be expanded upon: striatal activity

occurred during late adolescence/early adulthood, suggesting that this is an optimal period for learning." Why do these data suggest this is an optimal period for learning? Since the striatum is not the only (and arguably not most important) learning center, this is not an obvious leap to make.

The task participants were asked to learn was rather simple. Do the authors surmise that these effects would generalize to more complex tasks?

It is hard for this reviewer to see why distinguishing among the different subregions of the striatum is important. how does knowing this information help us better understand the mechanisms underlying adolescent learning?

Point-by-point Response:

Reviewer #2 (Remarks to the Author):

The goal of this study was to test the hypothesis that enhanced striatal activation in adolescents benefits learning performance. It is an extremely well-powered study (n=736 scans) that used a longitudinal design and the findings are interesting. I have very few comments, most of are clarifications or pertain to concerns raised in the previous round of review [redacted].

Methods

**How often was it the case that performance criterion was not reached (e.g. the participant got the full 12 trials before moving onto a new sequence)?
for the 'sensitivity to learning value' contrasts, did you exclude the sequence trials in which the participant did not learn (eg. they needed the full 12 trials before moving on)?**

To answer this question it is important to underline that we defined learning in a much more precise way than whether or not a participant needed the full 12 trials. That is, we defined successful learning per individual trial, rather than a combined aggregate over a sequence. This is described in the Methods section:

“To measure feedback learning performance, we calculated the percentage of trials in the learning phase for which feedback was successfully used on the next trial (stay for positive feedback, switch for negative feedback), relative to the total number of trials during the learning phase.”

In case participants completed 12 trials, this does not mean that learning did not take place. Stimuli were presented in a random order so it is possible that e.g. one stimulus did not appear for a couple of subsequent trials. We used this criterion-cutoff with the aim of striking a balance in the amount of trials during the learning compared to the application phase. To avoid confusion, we changed the term ‘performance criterion’ in the Methods section. This section now reads:

“After 12 trials, or when a performance criterion was reached (placing each of the three stimuli in the correct box at least two times). After that, a new sequence with three new stimuli was presented. This criterion was used to strike a balance between the number of trials in the learning and the application phase.”

To give a general overview, on average, participants needed 9.28 trials at T1 to reach the 12-trial limit. Of all the participants, there were only 3 who needed more than 11 trials on average per sequence. We now added the mean number of trials for each time point to the Methods section (page 13).

Since performance criterion was defined as placing each stimuli in the correct box twice, how do you know the second time in the correct place wasn't just by chance?

The percentage of times that participants provided an incorrect answer after an earlier correct answer indicating learning, was very low ($M=2.76\%$, $SD=2.21$ at T1), which supports the notion that learning truly occurred.

Given that there were three response options, the chance levels of choosing a correct option were 33% and for two subsequent correct sorts only 11% ($(1/3)^2$), which is relatively low.

Did these data control for IQ?

Feedback learning performance showed a weak age-controlled correlation with IQ (T1: $r=.140$, $p=.022$; T2: $r=.155$, $p=.019$). We now report this correlation with IQ in the Methods section.

Results: In response to my question about how the learning>application contrast yielded a quadratic trajectory whereas the positive>negative learning contrast did not, the authors have explained that these two contrasts measured two different processes. This needs to be clarified and explained in the text.

We have now clarified this further in the text by expanding our description of the two fMRI contrasts (page 4):

“We focused on the contrasts learning > application (sensitivity to learning value) and positive > negative learning (sensitivity to valence), which are two different processes which may have distinct developmental trajectories.”

Also, further discussion of why the relative peak in accumbens activity should be considered relevant when there is no difference between learning and application should be included.

To address this issue, we have added the following section to the Discussion:

“With regard to the different subregions in the striatum, we showed that for both dorsal and ventral caudate, there was an adolescent peak with increased activity for feedback with learning value compared to feedback without learning value. There was also an adolescent peak for nucleus accumbens with relatively more activity in this region for feedback with learning value compared to feedback without learning value.”

This discussion should include the references the authors provided to show that the nucleus accumbens can be blind to differences in valence, i.e. shows a similar response to

positive and negative feedback.

We agree that this is an interesting finding, and we have added the following section and the supporting reference to the Discussion:

“For nucleus accumbens, there was relatively more activity after positive feedback in childhood, whereas in early adulthood there were no longer valence differences in activity, suggesting that, similar to prior research in adults³¹, the nucleus accumbens can respond to cognitive learning signals regardless of whether they are presented as positive or negative feedback.”

Discussion

Some of the Discussion is a bit overstated in terms of causality. These data cannot tell us whether the unique adolescent brain patterns of activation lead to better learning, as this sentence suggests: "adolescent-specific sensitivities in brain development provide a window of opportunity for learning"

We agree that we cannot be sure yet of the direction of the effect. We have carefully checked the interpretation of our results to ensure that we are underlying the correlation between striatum activity and learning performance, and to ensure we are not assuming causality.

Relatedly, this sentence suggest big leaps that should be expanded upon: striatal activity occurred during late adolescence/early adulthood, suggesting that this is an optimal period for learning." Why do these data suggest this is an optimal period for learning? Since the striatum is not the only (and arguably not most important) learning center, this is not an obvious leap to make.

We have expanded upon this section to further support this claim. The behavioral results also suggest an adolescent peak in learning performance. Similarly, not only the striatum but also region in the prefrontal and parietal cortex (which are key regions in the learning network) showed an adolescent peak in neural activity. Consistent with several other studies showing increased cognitive performance in adolescence, the results provide convincing evidence for the hypothesis of an optimal learning period in adolescence. This section now reads (page 8):

“The peak in striatal activity for sensitivity to learning value occurred during late adolescence/early adulthood. Similarly, neural activity in the DLPFC, parietal cortex and SMA also showed a peak in activity around this age. Together with the behavioral results which demonstrated a peak in feedback learning performance in late adolescence/early adulthood, our data suggest that this developmental phase may be an optimal period for learning. Enhanced learning performance in adolescence has also been reported in prior studies^{18–20}.”

The task participants were asked to learn was rather simple. Do the authors surmise that these effects would generalize to more complex tasks?

This is an important point and at this moment, we cannot be sure of the answer yet. We have added this as a suggestion for future research to the Discussion section (page 9):

“For example, an important question concerns whether these findings are similar for more complex cognitive tasks, whether they are relevant for school learning and whether there are differences between supervised and unsupervised learning across development^{9,26}.”

It is hard for this reviewer to see why distinguishing among the different subregions of the striatum is important. how does knowing this information help us better understand the mechanisms underlying adolescent learning?

In prior studies, different subprocesses have often been ascribed to different subregions within the striatum (e.g. O’Doherty et al., 2004; Liljeholm et al., 2012; Garrison et al., 2013). Thus, it may be that if the underlying processes are different, there may also be differences in the developmental trajectories for these distinct subregions. The findings in this paper point to an adolescent peak in sensitivity to feedback with learning value compared to feedback without learning value in all subregions. In addition, only dorsal and ventral caudate activity predicted feedback learning performance. For sensitivity to valence, there were also differences in the developmental trajectories for activity for positive relative to negative feedback. Together, these findings show that the different subregions may have distinct developmental trajectories and this highlights the increased sensitivity of considering these regions separately. This is now also described in the Discussion (page 9-10).

References:

- O’Doherty, J. *et al.* Dissociable roles of ventral and dorsal striatum in instrumental conditioning. *Science*. **304**, 452–454 (2004).
- Liljeholm, M. & O’Doherty, J. P. Contributions of the striatum to learning, motivation, and performance: an associative account. *Trends Cogn. Sci.* **16**, 467–75 (2012).
- Garrison, J., Erdeniz, B. & Done, J. Prediction error in reinforcement learning: A meta analysis of neuroimaging studies. *Neurosci. Biobehav. Rev.* **37**, 1297–1310 (2013).